# Subject-Independent EEG Classification of Motor Imagery Based on Dual-Branch Feature Fusion

**DOI:** 10.3390/brainsci13071109

**Published:** 2023-07-21

**Authors:** Yanqing Dong, Xin Wen, Fang Gao, Chengxin Gao, Ruochen Cao, Jie Xiang, Rui Cao

**Affiliations:** 1School of Software, Taiyuan University of Technology, Taiyuan 030024, China; dongyanqing0078@link.tyut.edu.cn (Y.D.); cquwx0214@163.com (X.W.); volov628@outlook.com (F.G.); gaochengxin@tyut.edu.cn (C.G.); caory004@163.com (R.C.); 2College of Computer Science and Technology (College of Data Science), Taiyuan University of Technology, Taiyuan 030024, China; xiangjie@tyut.edu.cn

**Keywords:** motor imagery, subject-independent, deep learning, zero calibration, BCI

## Abstract

A brain computer interface (BCI) system helps people with motor dysfunction interact with the external environment. With the advancement of technology, BCI systems have been applied in practice, but their practicability and usability are still greatly challenged. A large amount of calibration time is often required before BCI systems are used, which can consume the patient’s energy and easily lead to anxiety. This paper proposes a novel motion-assisted method based on a novel dual-branch multiscale auto encoder network (MSAENet) to decode human brain motion imagery intentions, while introducing a central loss function to compensate for the shortcomings of traditional classifiers that only consider inter-class differences and ignore intra-class coupling. The effectiveness of the method is validated on three datasets, namely BCIIV2a, SMR-BCI and OpenBMI, to achieve zero calibration of the MI-BCI system. The results show that our proposed network displays good results on all three datasets. In the case of subject-independence, the MSAENet outperformed the other four comparison methods on the BCIIV2a and SMR-BCI datasets, while achieving F1_score values as high as 69.34% on the OpenBMI dataset. Our method maintains better classification accuracy with a small number of parameters and short prediction times, and the method achieves zero calibration of the MI-BCI system.

## 1. Introduction

A BCI system is a technology that establishes a connection between the human or animal brain and an external environment for information exchange and access control without the use of peripheral nerves and muscles, providing convenience for patients with limited motor function [1,2,3]. Due to the advantages of scalp electroencephalography (EEG) with high temporal resolution, simple operation, low cost and safety, BCI systems for EEG have received increasing attention from researchers [4,5]. Some common EEG paradigms include the event-related potential (ERP), P300, the steady-state visual evoked potential (SSVEP) and motor imagery (MI). MI means that the user imagines a specific part of the body in motion without external stimulation and self-induces to make the brain generate specific control signals [2,6]. MI-BCI converts EEG signals of subjects into control signals of external devices, which directly affects the clinical application of BCI and has great research value and potential. MI-BCI not only has a promising future in assisting disabled and medical rehabilitation, but also plays a role in entertainment, education and other fields [4].

The utility of the BCI system is inversely proportional to patient specificity, and a significant amount of calibration time is spent to capture the specificity of the current subject. The calibration time is generally about 20–30 min, which is time consuming in the actual application of BCI systems, depletes patient patience, increases patient anxiety and hinders the application of BCI in practice. Therefore, reducing the calibration time or even achieving zero calibration makes BCI plug-and-play a new research area in BCI application research. Kwon et al. achieved the first accuracy on subject independence over subject dependence through the selection of spatial–spectral features, driving the development of subject independence [5]. Zhang et al. adopted a pair of static filter bank common spatial patterns (OVR-FBCSP) for spatial feature pre-extraction, constructed a dual-branch network based on CNN and LSTM network for classification and trained an independent shared neural network using the EEG signals of all subjects as the training set [6]. Zhang et al. proposed an instance transfer learning based on a perceptual hashing algorithm to measure the similarity of spectrogram EEG signals among different subjects [7]. In addition, users vary in their thoughts, which may change the data distribution over time, leading to inherent signal non-stationarity properties of the EEG (due to power feature covariance bias), which affects the BCI performance [8,9,10]. Miladinović et al. applied stationary subspace analysis (SSA) to identify the most powerful spatial filtering method, which simulates real BCI scenarios to test how non-stationarity affects their performance, bridging the gap between the accuracy of BCI models observed in the calibration and test sets [11]. After these studies, it was found that the independent research on the subjects is still in the preliminary stage, which is far from the accuracy of subject dependence and also has certain limitations in practical applications. Therefore, finding a MI-BCI system with generalization ability has become a challenging area of scientific research.

Recently, with continuous research on motor imagery, feature extraction has developed from the previous single domain (time domain, frequency domain or spatial domain) to multi-domain fusion, especially in the fusion of spatial and frequency domain information. Discriminant feature representations can be extracted from raw EEG signals [5,12,13,14,15]. Bang et al. retained the variable structure and dependencies of space through spatial–spectral features, effectively weakening the differences between subjects, with the constructed network model having stronger robustness [13]. Cherloo et al. used the ensemble regularized common spatial spectrogram (Ensemble RCSSP) method to extract spatial spectral features, verifying that the proposed spatial–spectral discriminant representation can effectively distinguish motor imagery states [14]. In deep learning feature extraction, multi-branch model fusion has been shown to extract effective features in parallel, ensuring the diversity of discriminative features [16,17,18]. Yang et al. learned temporal and frequency domain features from EEG data in parallel through a two-branch CNN and used them as input to the MI decoder for classification [16]. Ma et al. used shallow dual-branch CNN for MI classification and demonstrated that the use of dual-branch parallel feature extraction was effective in improving classification accuracy [17]. Multi-scale feature extraction aims to learn latent features at different granularities through different receptive fields for EEG signals [19,20,21,22]. Zheng et al. utilized the MSFBCNN structure in the feature extraction layer to extract enough latent features to achieve feature diversification [22].

In this study, we propose an MSAENet to validate the performance on subject independence on three publicly available datasets, ultimately achieving zero calibration of the MI-BCI system. This novel neural network framework integrates AE with multi-scale deep learning models to learn EEG signal features from multiple aspects. To improve the generalization of this network, we pre-extracted EEG features of subjects in the frequency and spatial domains, reducing the interference of temporal information of subjects in different paradigms. In addition, we introduce a central loss function to compensate for the drawback that traditional Softmax only considers inter-class distance and ignores intra-class compactness.

## 2. Materials and Methods

### 2.1. Public Datasets

This study evaluates the generalizability of the MSAENet on three publicly available datasets: BCIIV2a [23], SMR-BCI [1,24,25] and OpenBMI [26].

The BCIIV2a dataset was obtained from 9 normal subjects, and the experiment consisted of 4 different classes of motor imagery tasks: left hand (class 1), right hand (class 2), feet (class 3) and tongue (class 4). At the beginning of each trial (t = 0), a fixed cross appeared on the computer screen and was accompanied by a brief siren sound. At t = 2 s, a left, right, down or up arrow (corresponding to the left hand, right hand, feet and tongue) appeared and stayed for 1.25 s, and the subject was asked to imagine the movement of the corresponding limb in the direction of the arrow until the arrow disappeared from the screen at t = 6 s, when the subject finished the imagery task. The entire experiment was performed using 22Ag/AgCl electrodes to acquire EEG signals at a sampling frequency of 250 Hz, and each subject performed a total of 288 trials. In this paper, the left- and right-handed EEG signals were used to achieve dichotomous task evaluation. Twenty EEG channels in motor cortical regions (FC3, FC1, C1, FCz, FC2, FC4, C5, C3, C1, Cz, C2, C4, C6, CP3, CP1, CPz, CP4, P1, Pz, and P2) were selected from the raw EEG signals, and the 3 s–6 s time period was intercepted as EEG-MI data. Subjects continued to perform the motor imagery task for a selected period of 3 s.

The SMR-BCI dataset is derived from 14 healthy subjects, and the experiment includes right hand and feet motor imagery binary classification tasks. A sampling frequency of 512 Hz was used to acquire 15 channels of EEG signals. In this paper, the sampling frequency of 512 Hz was downsampled to 250 Hz using the first 4 s of continuous motion imagery data.

The OpenBMI dataset comes from 54 healthy subjects, and the experiment records dichotomous tasks for left and right hand motor imagery. From 0 to 3 s of each trial, a black fixed cross appeared in the center of the monitor to prepare the subjects for the MI task. When an arrow to the left or right appeared, the subjects performed the corresponding imagery task with a continuous grasp for 4 s. The experiment consisted of 4 phases, each with 100 trials, including 50 trials each for the imaginary left-handed and right-handed tasks. A sampling frequency of 1000 Hz was used to acquire 62 channels of EEG signals. In this paper, we selected 20 electrode channels from the motor cortex region. EEG signals were processed, the 1000 Hz sampling frequency was down-sampled to 250 Hz, and the 4 s EEG signals for motor imagery grasping were intercepted.

### 2.2. Spatial–Spectral Feature Pre-Extraction

All three datasets used in this study were EEG signals recorded according to a time series, and the time-domain signals that varied over time during cross-session and cross-subject motor imagery classification made the features unstable [27]. Therefore, we adopted a feature pre-extraction method to solve this problem. Motor imagery showed obvious performance in the μ and β rhythms, and the selected characteristic frequency bands were mainly distributed between 8–30 Hz [26]. We define $X_{N}=\left\{x_{i}\right\}\;(i=1,2,...,n)\in R_{T\times C}$ as the experimentally obtained raw EEG signal; $Y_{N}=\left\{y_{i}\right\}\;(i=1,2,...,n)$ is the corresponding category, where N denotes the number of EEG trials, T denotes the number of sampling points, and C denotes the number of EEG channels. The application of bandpass filtering is as follows:(1)$$E_{N}=B_{N}⊗X_{N}$$

$E_{N}$ is the data obtained after bandpass filtering from 8–30 Hz, where $X_{N}$ is the original EEG signal, ⊗ is defined as a fifth-order Butterworth bandpass filter operation, and$\;B_{N}$ is a frequency band at a specific frequency.

The spatial–spectral features are pre-extracted from the perspective of spectral and spatial optimization. After bandpass filtering based on the 8–30 Hz band, the CSP algorithm is used for processing [28,29,30,31], where $W_{N}\in R_{U\times U}$ is the feature matrix containing spatial feature information obtained from the $E_{N}$ resolution, and U is the number of spatial filters obtained by the CSP algorithm. $E_{N}$ contains the spectral information extracted from a specific frequency band, and the spatial–spectral features are obtained using the covariance matrix. The covariance matrix is as follows:(2)$$C_{p}=\mathrm{cov}({W_{N}}^{T}•E_{N})$$

### 2.3. Classification

Our proposed MSAENet structure is shown in Figure 1, which consists of three modules: the autoencoder branch (AE branch), the multiscale branch and the loss function fusion (LFF). We conduct subject-independent and subject-dependent experiments on three publicly available datasets to demonstrate the strong generalization of the MSAENet and consider the performance of all methods using two evaluation metrics: accuracy (ACC) and F1_score. The subject-leave-one-cross validation method is used for the subject-independent case, and, for a particular dataset containing S subjects, the session data of a subject are used as the test set. The data of the remaining S-1 subjects are used as the training set. The method uses the Keras framework (TensorFlowv2.2.0 as the backend), and the training process is implemented using NVIDIA GeForce RTX 2080 Ti with 22 GB of memory. During the training iterations, the Adam optimizer is used to optimize the loss function, and the early stop method is used for network training. The learning rate decay is 0.5 when there is no improvement in validation loss for 5 consecutive epochs, and the network stops the iteration process when there is no reduction in the validation loss for 20 consecutive epochs.

#### 2.3.1. AE Branch

The AE branch is mainly composed of two components: encoder E=p(Input) and decoder D=q(z). The encoder encodes high-dimensional data into a one-dimensional latent vector z that can maintain the robustness characteristics of the EEG-MI signal by convolutional pooling, and the whole encoder is considered as a data dimensionality reduction process; the detailed structure is shown in Table 1. Average pooling achieves weight sharing, effectively reducing the number of parameters. The BN layer speeds up network training, convergence and prevents network overfitting.

The decoder and the encoder have a symmetric structure. First, the latent vector z is amplified to the appropriate size by a dense layer operation, and the one-dimensional vector is decoded into reconstruction which has the same data size as input from two transposed convolutional layers. Transposed convolution can extract effective features from the data, reduce useless features and facilitate the reconstruction of latent vectors.

#### 2.3.2. Multiscale Branch

The multiscale branch uses different sub-branches to extract spatial–spectral features at different granularities and uses three different convolution kernels of 3 × 3, 5 × 5 and 7 × 7 to extract local features from the spatial–spectral data. Different granularity of convolution kernels is used to extract local features and to ensure that richer and more representative features are extracted. Then, the convolution is carried out through a 3 × 3 convolution kernel. With the deepening of the network, the extracted local features are also richer so as to achieve a wider EEG receptive field. See Table 2 for details.

#### 2.3.3. Loss Function Fusion

In this study, three loss functions are used to jointly constrain the training of the whole network. The first loss function is the mean squared error (MSE) loss function. The AE branch trains this branch by minimizing the reconstruction error between the encoder input data input and the decoder output data reconstruction. The intermediate latent vector z can maintain the features in the input data. Given an input signal $C_{p}=\left\{\begin{array}{cccc} x_{11} & x_{12} & \cdots & x_{1C}\\ x_{21} & x_{22} & \cdots & x_{2C}\\ \vdots & \vdots & \ddots & \vdots \\ x_{C1} & x_{C2} & \cdots & x_{CC} \end{array}\right\}$, its loss function is expressed as follows:(3)$$L_{MSE}(x,\overset{̑}{x})=\frac{1}{C^{2}}\sum_{i=1}^{C}\sum_{j=1}^{C}{\left\|x_{ij}-{\overset{̑}{x}}_{ij}\right\|}^{2}$$
where ${\overset{̑}{x}}_{ij}$ is the value of the matrix in the reconstructed data.

In the MSAENet classification module, Softmax is used as a classifier, and the spliced feature vector is activated by Softmax and input to the fully connected layer. The weight coefficient of each class is obtained, which is expressed as follows:(4)$$\overset{̑}{y}\left(m\;,\;z\right)=Soft\max\left(W(m+z)+b\right)$$
where W and b are the weight matrix and bias vector, respectively. Then, the model is trained using the Adam optimizer to minimize the cross-entropy loss, which makes the spacing between two categories larger and can effectively distinguish two different categories to achieve a good classification effect. The calculation method is as follows:(5)$$L_{cross-entropy}(y,\overset{̑}{y})=-\sum_{k=1}^{M}y_{k}{\overset{̑}{\log y}}_{k}$$
where y and $\hat{y}$ are the true label and classification probability, respectively. M indicates the number of categories.

Then, the central loss function is used to effectively make up for the defects of the Softmax layer in the traditional CNN, not only considering the dispersion between different categories but also the cohesion of the same category, thus making the intra-class distance smaller and maintaining intra-class compactness and central loss. The function is expressed as follows:(6)$$L_{center-loss}(m_{i},z_{i})=\frac{1}{2}\sum_{i=1}^{b}{\left\|m_{i}+z_{i}-c_{y}\right\|}_{2}^{2}$$

Among them, $c_{y}$ represents the feature center of the yth category, $m_{i}$ represents the feature before the fully connected layer of the multi scale branch, $z_{i}$ represents the feature of the fully connected layer of the AE branch network, and b represents the number of batches. It is hoped that the smaller the sum of the squares of the distance between the feature and the feature center of each sample in each batch, the better.

The MSAENet was optimized by combining the mean square error loss function in Equation (3), the cross entropy loss function in Equation (5) and the central loss function in Equation (6). The final loss function is expressed as follows:(7)$$L\left(x,\overset{̑}{x},y,\overset{̑}{y},m_{i},z_{i}\right)=\beta_{1}L_{MSE}\left(x,\overset{̑}{x}\right)+\beta_{2}L_{cross-entropy}\left(y,\overset{̑}{y}\right)+\beta_{3}L_{center-loss}\left(m_{i},z_{i}\right)$$

$\beta_{1}$, $\beta_{2}$ and $\beta_{3}$ are used to denote the weighted hyperparameters of each loss function, and the network is trained by back-propagating the error through the integration of the three loss functions. We found that the accuracy is higher when the loss weight of the cross loss function is at its maximum in reference [24]. In this paper, we choose (0.1, 1.0, 0.1) (0.5, 1.0, 0.5) (0.5, 0.5, 0.5) (1.0, 1.0, 1.0) (1.0, 0.5, 1.0) groups of hyperparameters of loss function weights for testing, and we find that the accuracy is higher when $\beta_{1}$ = 0.5, $\beta_{2}\;$= 1.0 and $\beta_{3}\;$= 0.5. We observe that a larger classification loss weight for Softmax results in slightly higher classification accuracy, a result that is more consistent with those in the relevant study [24].

### 2.4. Comparative Experiment

#### 2.4.1. EEGNet

The EEGNet is a lightweight CNN framework for classification in different BCI paradigms. First, a 1 × 200 convolution kernel is used to extract the temporal information from the original data EEG. Then, the spatial features are extracted from the n channels using an n × 1 convolution kernel. Finally, the separable convolution summarizes each feature map separately and optimally combines the outputs, which can effectively reduce the number of fitting parameters [32].

#### 2.4.2. DeepConvNet

The DeepConvNet proved to be effective in handling EEG-MI classification. The network consists of 4 blocks: the first block contains two convolutional layers and BatchNormalization, using the maximum pooling layer to reduce the number of parameters; the final dropout layer prevents overfitting. The last three blocks have the same structure, with the only difference from the first block being that there is only one convolutional layer. Band-pass filtering (fifth-order Butterworth band-pass filter) is performed on the original EEG signal between 8 and 30 Hz [33].

#### 2.4.3. MIN2Net

The MIN2Net adopts an autoencoder network combined with a momentum learning mechanism and demonstrates advanced performance in subject independence. The framework consists of an encoder and a decoder. Different linear convolution kernels are used in the encoder to extract spatial–temporal features, and the autoencoder is back-propagated through the momentum learning mechanism and the error of the reconstruction function [24].

#### 2.4.4. SSCNN

Compared with subject dependence, the SSCNN shows stronger advantages in subject independence. The network obtains spatial–spectral features by extracting the spatial features of 30 frequency bands, selects the spatial–spectral features of the first 20 frequency bands through mutual information, sends them to a three-layer convolutional network, and, finally, splices the classification through the fully connected layer [5].

### 2.5. Ablation Method

To prove the validity of the dual branch, we only keep the first branch of the dual branch, which extracts spatial–spectral features at different granularities through different receptive fields, and we define the network as the MultiScaleNet; the second branch is the AE, which learns the features of the input data as much as possible through the potential vector z in between. We define it as the AENet.

To test the effectiveness of multiple scales in the network structure, 3 × 3 convolutional kernels are used instead of different scale convolutions in all the proposed networks, which are defined as the SingleScaleNet.

The central loss function can increase the compactness within the class. This study introduces the central loss function to make up for the deficiencies in the Softmax loss function. It removes the central loss function to verify its validity. The network structure is called the WithoutLossNet.

To demonstrate the effectiveness of feature pre-extraction, the raw data are directly used as the input data of the MSAENet, which is called the RawDataNet.

### 2.6. Visisual

Deep learning is a black-box model, and information about the intermediate processes is not directly observable to us. To visualize the features of deep learning, we use the t-SNE method [34] to embed the fully connected layer of the deep learning model into a two-dimensional space for visualization and compare the differences in the features learned by various methods.

## 3. Results

The experimental results of the subject-independent comparison experiments on the three publicly available datasets are shown in Table 3. On both the BCIIV2a and SMR_BCI datasets, the MSAENet is significantly higher than the other four advanced comparison methods in two evaluation metrics, i.e., accuracy and F1_score, with an accuracy rate of 72.50% and an F1_score of 72.23% on SMR-BCI. However, in the OpenBMI dataset, MIN2Net has the best performance with the highest values on both the ACC and F1_score metrics, while the EEGNet, the DeepConvNet, the SSCNN and the MSAENet have comparable results. The F1-Score, as a comprehensive indicator, balances the impact of precision and recall. Using just one of the two indicators of the value of a large indicator is not useful; it must be the value of the two indicators at the same time. The corresponding value of the F1 Score indicator will be very large, which can be a more comprehensive evaluation of a classifier. The EEGNet network has the worst results on the SMR-BCI dataset with an F1_score value of only 34.43%. The DeepConvNet has a low ACC on the BCIIV2a dataset; the F1_score value, in particular, only reaches 30.62%. This situation is similar to that of the EEGNet, with neither of them being generalizable. The MIN2Net network in the BCIIV2a dataset has an F1_score value of 49.09%. The results of SSCNN and MSAENet on the three datasets show robustness, but the results of the MSAENet are higher than those of the SSCNN.

MSAENet embodies great advantages over the other four methods compared, and the ablation experiment is used to determine the degree of influence of various factors on the network. The results of the subject independence in the ablation experiments are shown in Table 4. It can be seen that the MSAENet outperforms the other ablation network results in both accuracy and F1-score in the subject-independent case. In particular, compared with the RawDataNet, our method shows great advantages on two datasets, i.e., BCIIV2a and SMR-BCI, with significant results in F1_score and ACC. In the OpenBMI dataset, the performance of the AENet network is poor, but the overall trend of several ablation experiments has not changed significantly. On the BCIIVa dataset, the SingleScaleNet differs the least from the MSAENet in terms of ACC, and the WithoutLossNet is closest to the MSAENet in terms of the F1_score. On the SMR-BCI dataset, the results of MultiScaleNet are close to those of MSAENet; this was the smallest difference.

## 4. Discussion

### 4.1. Comparative Test Analysis

From the results of the subject-independent comparison experiments in Table 3, it can be seen that the MSAENet shows the best performance on both the BCIIV2a and the SMR-BCI datasets. On the OpenBMI dataset, the MIN2Net shows significantly higher results than the other four methods, but there is no significant difference among the results of the other four methods. The EEGNet is a lightweight and compact framework for learning potential features by deep convolution and separable convolution, and the EEGNet contains temporal convolution kernels in the first block, which are half the length of the sampling rate, for extracting time-domain information. However, the input data distribution of EEG-based BCI systems may change during intra-session transitions [11], and the time-domain features are variable, which cannot be easily captured as stable features when considering feature extraction between different sessions; thus, the EEGNet may not show good generalizability when applied to multiple datasets because of the non-stationary characteristics of the EEG signals. Compared with the EEGNet, the MSAENet can learn features more comprehensively in the spectral and spatial domains, capture features at different scales and extract differences between sessions, which is more suitable for subject independence and reduces the calibration time of subjects. The DeepConvNet network shows a trend consistent with EEGNet, with a low F1_score and poor generalization on the BCIIV2a dataset. It has been mentioned in the literature that F1_score is more valuable than ACC [35].The DeepConvNet learns potential features of subjects from deep level by four blocks; due to the paradigm of different datasets, it may involve different time-domain, frequency domain and spatial domain information, but the DeepConvNet has not considered multi-scale and multi-domain fusion up to this point. Therefore, the DeepConvNet has some advantages for a single dataset but cannot show stable reliability on every dataset when multiple datasets are considered. Compared to the DeepConvNet, the MSAENet performs multi-domain fusion from spectral and spatial domain aspects to capture the unique features of different domains. The MIN2Net shows an advantage on the OpenBMI dataset but does not show a great advantage on the other two datasets, especially the dataset BCIIV2a with an F1_score of only 49.09%; this differs from the MSAENet by 19.81%. Although the MSAENet does not show the best performance on the OpenBMI dataset as it only differs from the optimal result by 3.28%, which means that it has strong generalizability. Our analysis of the MIN2Net’s strong generalizability is that it is due to the integration of deep metric learning into the autoencoder to learn compact and discriminative potential features from the EEG and to classify them simultaneously. While the MSAENet can maintain better generalization with the MIN2Net, in addition to extracting features from multiple scales and domains, we also improve Softmax’s problem that it only considers inter-class distance and ignores intra-class compactness to learn richer latent features from EEG. The SSCNN also shows better stability on the SSCNN based on spectral and spatial domain feature fusion, but it fails to refine the feature granularity from multiple scales. The MSAENet integrates multi-domain fusion, refines feature extraction granularity and learns intra-class compact features and richer feature representations from different sessions, thus achieving satisfactory results in subject independence with strong generalizability.

### 4.2. Analysis of Ablation Experiments

Through comparative experiments, we found that the MSAENet showed great superiority compared with the other four methods. The advantages of the MSAENet were further analyzed using ablation experiments. From the results of the subject-independent ablation experiments in Table 4, we found that the RawDataNet network without spatial–spectral feature pre-extraction had the greatest difference with the MSAENet, and its results were much lower than those of the MSAENet, especially on the two datasets, i.e., BCIIV2a and SMR-BCI; the maximum difference in accuracy is about 20%. The results show that the spatial–spectral feature pre-extraction can abstract the spatial–spectral signals of multiple subjects into one dimension and reduce the network’s training time and its difficulty. It can not only solve the differences between subjects and different datasets but also retain the original data characteristics as much as possible and eliminate invalid information. Spatial–spectral features combined with CSP and frequency bands can be used as a feature extraction method for motor imagery learning and classification [14]. Both the MultiScaleNet and the AENet can prove that multi-branch network models can learn features from different angles; that different convolution methods have different feature extraction capabilities; and that the multi-branch can realize feature extraction diversification [17]. The WithoutLossNet proves the effectiveness of the central loss function on the BCIIV2a and SMR-BCI datasets. The central loss function can well compensate for the shortcomings of Softmax. The combination of the central loss function and the Softmax classifier can consider the relationship between the class and different classes at the same time, so that the same type of data converges to the feature center, and the distance between the feature centers between different classes expands [36,37]. The SingleScaleNet replaces different scale convolution kernels with 3 × 3 convolution kernels, which verifies that the features extracted from different scales of sensory fields are diverse. The feature representations learned by multi-scale are more comprehensive and diverse. Overall, our network integrates the advantages of different components, and the learned features are more diverse and comprehensive.

### 4.3. Parameter Performance Analysis

The results for the number of network parameters and prediction time for the comparison experiments are shown in Table 5. Although the number of parameters in the EEGNet and the DeepConvNet is much lower than that of the MIN2Net, the SSCNN and the MSAENet, these two methods do not have applicability. The number of training parameters of the SSCNN is as high as 77,577,714, and the average prediction time is as long as 1.0934 s on OpenBMI. While MSAENet keeps the two evaluation indicators of ACC and F1_score higher than SSCNN, its trainable parameters and prediction time are much lower than the SSCNN. The MSAENet greatly reduces the number of parameters and shortens the prediction time while maintaining a high accuracy rate.

### 4.4. Visualization Analysis

The visualization results of the comparison experiment with the subject-independent setting are shown in Figure 2. The one-dimensional vector of the last layer of the MSAENet is embedded into a two-dimensional space for visualization by the t-SNE method. Different shapes indicate that different parts of the motor imagery are performed: triangles indicate the left hand, circles indicate the feet, and crosses indicate the right hand. Different colors are used to distinguish the visualization results in order to represent them more intuitively. The MSAENet’s visualization results in better classification than other methods, with a more compact intra-class distribution and larger inter-class spacing. The visualization distribution of the MSAENet outperforms the other networks on the two datasets BCIIV2a and SMR-BCI, which is consistent with the results of the data in Table 3.

The visualization results of the subject-independent ablation experiment are shown in Figure 3. It can be seen from the figure that in the RawDataNet network the visualization results of the three datasets are messy and cannot effectively distinguish the imaginary tasks. Compared with other ablation experiments, the MSAENet not only has high classification accuracy but also more compact intra-class distance and good cohesion, which further confirms the data in Table 4.

### 4.5. Future Directions

Despite the good results of the MSAENet on subject independence, there are still some shortcomings that need further improvement. First, the attention mechanism module can automatically extract the features of interest. The integration of the attention mechanism module into the MSAENet module to further improve classification performance should be considered. Second, the MSAENet extracted valid discriminative features, which were considered for application in other EEG measurements, such as SSVEP. Finally, combining the MSAENet with transfer learning to improve the generalization of the BCI system can be considered.

## 5. Conclusions

In this paper, a two-branch network structure based on feature fusion is proposed. It was verified that AE and multi-scale feature fusion can extract richer features from EEG signals. A feature pre-extraction approach for data based on both the spatial and frequency domains is expected to extract common information of subjects from different paradigms. The introduction of the centroid function can compensate for the shortcoming of traditional Softmax, which ignores intra-class cohesiveness. Our proposed network shows superior performance on all three datasets, demonstrating that the network is highly generalizable, achieves zero calibration and facilitates the development of applications for the MI-BCI system.

## Figures and Tables

**Figure 1 brainsci-13-01109-f001:**
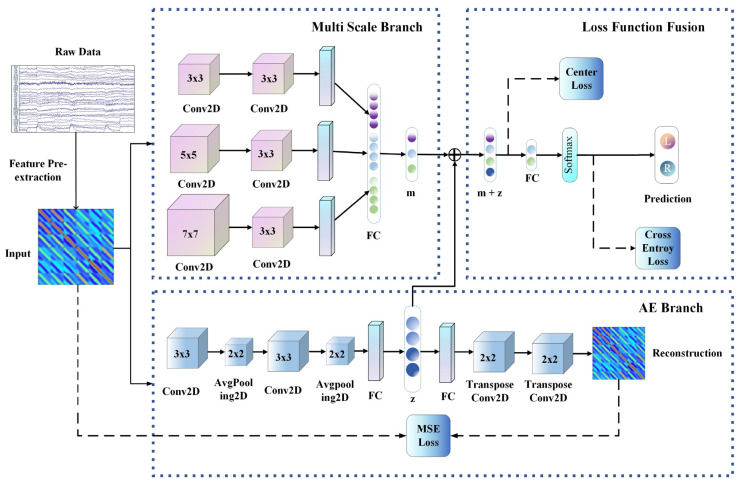
MSAENet structure. RawData refers to EEG data downsampled to 250 Hz. Input refers to the spatial–spectral features obtained through the common spatial mode and the 8–30 Hz frequency band. Multi scale branch refers to feature extraction through convolution kernels of different sizes. The AE branch is composed of two parts: the encoder and the decoder, and the intermediate vector z is adjusted by training the error between the input data and the reconstructed data together with other loss functions. Loss function fusion refers to the combination of the central loss function, the mean square error loss function and the cross entropy loss function to backpropagate the training network.

**Figure 2 brainsci-13-01109-f002:**
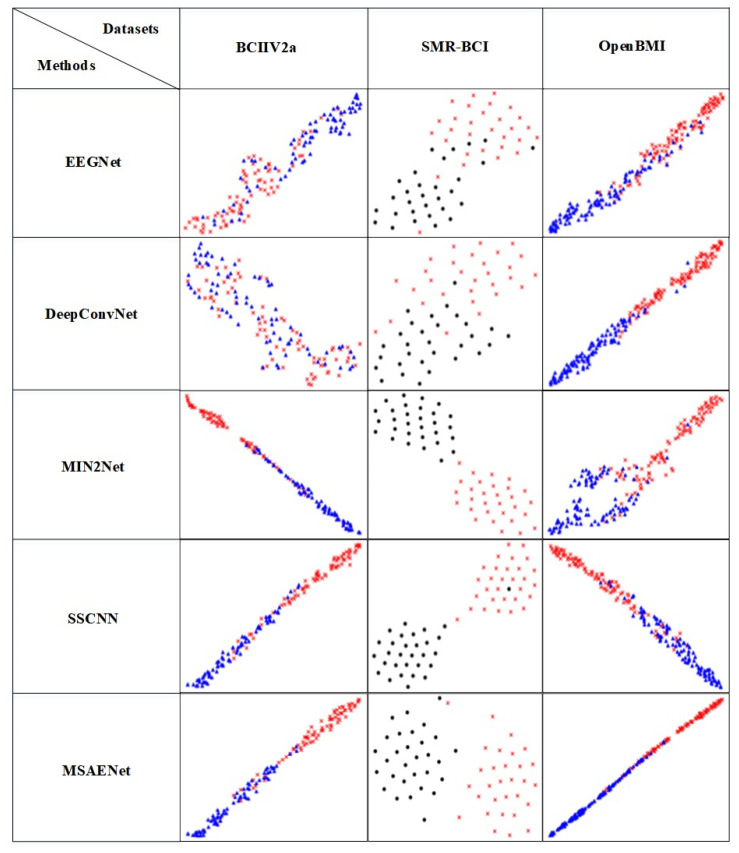
Visualization results of subject-independent comparison experiments. The visualization results are obtained by t-SNE embedding the last layer in the network in two dimensions. 
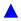
 represent left hand MI, 
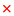
 represent right hand MI, 
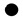
 represent feet MI.

**Figure 3 brainsci-13-01109-f003:**
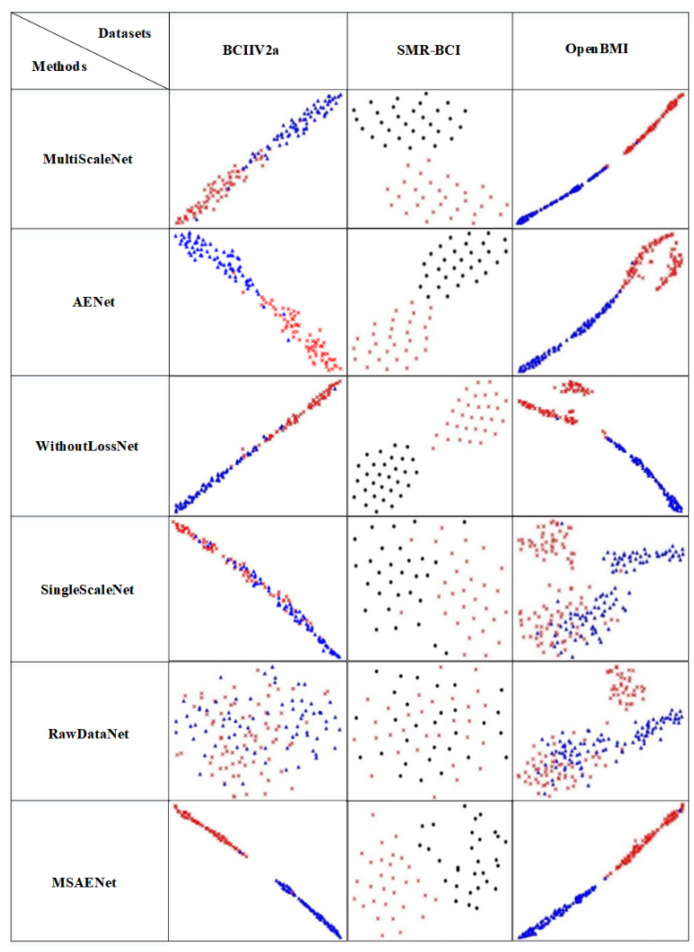
Visualization results of subject-independent ablation experiments. The visualization results are obtained by t-SNE embedding the last layer in the network in two dimensions. 
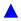
 represent left hand MI, 
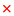
 represent right hand MI, 
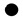
 represent feet MI.

**Table 1 brainsci-13-01109-t001:** AE branch network architecture.

Blocks	Layer	Filter	Size	Stride	Activation	Options	Output
Encoder	Input		(28,28,1)				(28,28,1)
	Conv2D	1	(3,3)	(1,1)	ReLU	padding = same	(28,28,1)
	BatchNormalization						(28,28,1)
	AveragePooling2D		(2,2)	(1,1)			(14,14,1)
	Conv2D	10	(3,3)	(1,1)	ReLU	Padding = same	(14,14,10)
	BatchNormalization						(14,14,10)
	AveragePooling2D		(2,2)	(1,1)			(7,7,10)
	Flatten						(490)
Latent	FC(z)		(20)				(20)
Decoder	Dense		(490)				(490)
	Reshape		(7,7,10)		ReLU		(7,7,10)
	Conv2DTranspose	10	(14,14,10)	(2,2)	ReLU	Padding = same	(14,14,10)
	Conv2DTranspose	1	(28,28,1)	(2,2)	ReLU	Padding = same	(28,28,1)

**Table 2 brainsci-13-01109-t002:** Multi scale branch network architecture.

Blocks	Layer	Filter	Size	Stride	Activation	Options	Output
Multi scale branch	Input		(28,28,1)				(28,28,1)
	Conv2D	10	(3,3)	(1,1)	ReLU	padding = same	(28,28,10)
		10	(5,5)	(1,1)	ReLU	padding = same	(28,28,10)
		10	(7,7)	(1,1)	ReLU	padding = same	(28,28,10)
	Conv2D	20	(3,3)	(1,1)	ReLU	Padding = same	(28,28,20)
		20	(3,3)	(1,1)	ReLU	Padding = same	(28,28,20)
		20	(3,3)	(1,1)	ReLU	Padding = same	(28,28,20)
	Concatenate						(28,28,60)
	Flatten						(15360)
	Dense		(256)				(256)
	Dense(m)		(20)				(20)
Feature fusion	Add		(20)			m + z	(20)
Classification	Dense	2			Softmax		2

**Table 3 brainsci-13-01109-t003:** Results of subject-independent comparison experiments ($\beta_{1}$ = 0.5, $\beta_{2}$ = 1.0, $\beta_{3}$ = 0.5). The largest value is marked with bold font.

	BCIIV2a		SMR-BCI		OpenBMI	
	ACC	F1_Score	ACC	F1_Score	ACC	F1_Score
EEGNet	64.26 ± 11.03	60.19 ± 19.96	58.07 ± 11.45	34.43 ± 31.35	68.84 ± 13.87	70.39 ± 14.30
DeepConvNet	56.34 ± 8.86	30.62 ± 28.96	65.26 ± 16.83	54.38 ± 32.58	68.33 ± 15.33	70.20 ± 15.18
MIN2Net	60.03 ± 9.24	49.09 ± 23.28	59.79 ± 13.72	61.10 ± 23.64	**72.03 ± 14.04**	**72.62 ± 14.14**
SSCNN	66.05 ± 13.70	61.91 ± 20.31	66.21 ± 15.15	54.36 ± 31.21	68.27 ± 13.56	65.86 ± 17.37
MSAENet	**69.98 ± 12.15**	**68.90 ± 11.77**	**72.50 ± 13.58**	**72.23 ± 14.18**	68.31 ± 14.28	69.34 ± 13.75

**Table 4 brainsci-13-01109-t004:** Results of subject-independent ablation experiments. The largest value is marked with bold font.

	BCIIV2a		SMR-BCI		OpenBMI	
	ACC	F1_Score	ACC	F1_Score	ACC	F1_Score
MultiScaleNet	64.66 ± 17.38	56.48 ± 30.29	69.05 ± 14.95	64.59 ± 25.85	67.23 ± 14.13	68.06 ± 14.79
AENet	66.36 ± 13.64	65.07 ± 14.74	60.95 ± 14.36	57.45 ± 24.70	66.53 ± 12.63	62.90 ± 16.65
WithoutLossNet	66.28 ± 17.33	67.30 ± 17.03	68.45 ± 17.36	63.82 ± 27.38	67.96 ± 14.03	66.08 ± 16.32
SingleScaleNet	68.29 ± 14.70	65.27 ± 17.92	66.19 ± 14.36	60.64 ± 26.09	67.44 ± 13.73	65.91 ± 16.31
RawDataNet	55.56 ± 4.50	55.32 ± 4.74	56.08 ± 6.15	50.35 ± 15.92	67.80 ± 11.80	68.05 ± 12.08
MSAENet	**69.98 ± 12.15**	**68.90 ± 11.77**	**72.50 ± 13.58**	**72.23 ± 14.18**	**68.31 ± 14.28**	**69.34 ± 13.75**

**Table 5 brainsci-13-01109-t005:** Network parameters.

	Comparison Model	Trainable Params	Subject-Dependent	Subject-Independent
			Prediction Time	Prediction Time
BCIIV2a	EEGNet	5162	0.1173	0.0920
	DeepConvNet	151,027	0.1617	0.1739
	MIN2Net	55,232	0.1803	0.2373
	SSCNN	77,577,714	0.7600	0.7444
	MSAENet	12,053,956	0.2334	0.2602
SMR-BCI	EEGNet	5082	0.1296	0.1105
	DeepConvNet	150,302	0.1519	0.1906
	MIN2Net	38,297	0.2433	0.2966
	SSCNN	54,076,514	1.0257	0.6688
	MSAENet	3,958,623	0.4079	0.4375
OpenBMI	EEGNet	5162	0.1439	0.1372
	DeepConvNet	151,027	0.1618	0.4735
	MIN2Net	55,232	0.2851	0.1043
	SSCNN	77,577,714	1.0934	0.8560
	MSAENet	12,053,956	0.2541	0.2445

## Data Availability

BCIIV2a Dataset at BCI Competition IV (bbci.de).

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
