# Peer review of "Subject-Independent EEG Classification of Motor Imagery Based on Dual-Branch Feature Fusion"

_brainsci, 2023, doi:10.3390/brainsci13071109_

Round 1

Reviewer 1 Report

Comments and Suggestions for Authors

1. The author should provide a proper explanation for the choice of hyperparameters.

2. There is no mention or clarification about the activation functions used, what is ELU and RELU?

3. It is necessary to discuss why MSAENet outperforms other methods, even though deep learning is considered a black-box approach.

4. How does the proposed method's performance compare to previous works?

Author Response

Thank you very much for taking time out of your busy schedule to review my work, I will explain and revise the issues you have pointed out point by point, I am uploading it as an attachment "reviewer1.doxc", the highlighted parts are the parts that I have revised in the manuscript. Please see the attachment. Thank you very much for your comments and approval of my work!

Reviewer 2 Report

Comments and Suggestions for Authors

Title: Review of "Subject-independent EEG Classification of Motor Imagery based on a Dual-branch Feature Fusion"

General Comments:

The paper presents a novel approach to address the challenges faced by Brain Computer Interface (BCI) systems in terms of practicability and usability. The proposed method utilizes a dual-branch Multi Scale Auto Encoder network (MSAENet) combined with a central loss function to decode human brain motion imagery intentions. The authors validate the effectiveness of their approach on three datasets and compare it with four other methods. The results demonstrate that the proposed network achieves superior performance in subject-independent EEG classification of motor imagery, while maintaining low computational complexity and zero calibration of the MI-BCI system.

Strengths:

Originality: The paper introduces a new method that combines a dual-branch MSAENet with a central loss function, providing a novel approach to decode motor imagery intentions from EEG signals.

Suggestions for Improvement:

"Common EEG events are event-related potential (ERP), P300, steady-state visual evoked potential (SSVEP), and motor imagery (MI)."

I suggest that the authors change the word events to paradigms. 

"The utility of the BCI system is inversely proportional to patient specificity, and a  significant amount of calibration time is spent to capture the specificity of the current subject. The calibration time is generally about 20-30 minutes, which is time-consuming in the actual application of BCI systems"

I appreciate the authors for addressing the issue. It not only time-consuming for the patients, but also there is a problem that EEG changes over time, which is often overlooked. I would suggest that the authors expand their literature review to encompass the topic of nonstationarity or temporal changes (even within 20-30min, from calibration to the online application of BCI). These factors can significantly impact the feature extraction process, resulting in non-stationary feature covariance shifts. This concern has been discussed in a relevant study (10.1016/j.cmpb.2020.105808) and should be taken into consideration.

While the presented results are promising, it is essential for the authors to discuss potential issues. Precisely, the extraction of features considering the entire 3 seconds of motor imagery may not align with the time-critical nature of BCI systems that involve human interaction. The delay of 3 seconds between the onset of motor imagery and obtaining a response is typically unacceptable for users, as it affects their perception of being in control. Therefore, it would be valuable for the authors to address this issue and propose ideas to improve the time-critical perspective.

Furthermore, it would be beneficial to compare their system with more traditional approaches mentioned in the introduction, especially in terms of dealing with noisy and nonstationary EEG data.

Overall, the work is a well-executed study that, with the mentioned improvements, could contribute to the advancement of brain-computer interface technology.

Author Response

Thank you very much for taking time out of your busy schedule to review my work, I will explain and revise the issues you have pointed out point by point, I am uploading it as an attachment "reviewer 2.doxc", the highlighted parts are the parts that I have revised in the manuscript. Please see the attachment. Thank you very much for your comments and approval of my work!

Round 2

Reviewer 2 Report

Comments and Suggestions for Authors

The Authors have addressed all of my concerns with the original manuscript. The revised manuscript is ready for publication.